# Presymptomatic microRNA-based biomarker signatures for the prognosis of localized radiation injury in mice

Lucie Ancel[1], Jules Gueguen[1], Guillaume Thoër[1], Jules Marçais[1], Aïda Chemloul[1], Bernard Le Guen[2], Marc Benderitter[3], Radia Tamarat[3], Maâmar Souidi[1], Mohamed Amine Benadjaoud[1], Stéphane Flamant[1]*

**1** Autorité de Sûreté Nucléaire et de Radioprotection (ASNR), PSE-SANTE/SERAMED/LRAcc, Fontenay-aux-Roses, France, **2** EDF, DPN, 1 place Pleyel, Saint Denis, France, **3** ASNR, PSE-SANTE, Fontenay-aux-Roses, France

☉ These authors contributed equally to this work.
* stephane.flamant@asnr.fr

## Abstract

The threat of nuclear or radiological events requires early diagnostic tools for radiation induced health effects. Localized radiation injuries (LRI) are severe outcomes of such events, characterized by a latent presymptomatic phase followed by symptom onset ranging from erythema and edema to ulceration and tissue necrosis. Early diagnosis is crucial for effective triage and adapted treatment, potentially through minimally invasive biomarkers including circulating microRNAs (miRNAs), which have been correlated with tissue injuries and radiation exposure, suggesting their potential in diagnosing LRI. In this study, we sought to identify early miRNA signatures for LRI severity prognosis before clinical symptoms appear. Using a mouse model of hindlimb irradiation at 0, 20, 40, or 80 Gy previously shown to lead to localized injuries of different severities, we performed broad-spectrum plasma miRNA profiling at two latency stages (day 1 and 7 post-irradiation). The identified candidate miRNAs were then challenged using two independent mouse cohorts to refine miRNA signatures. Through sparse partial least square discriminant analysis (sPLS-DA), signatures of 14 and 16 plasma miRNAs segregated animals according to dose groups at day 1 and day 7, respectively. Interestingly, these signatures shared 9 miRNAs, including miR-19a-3p, miR-93-5p, miR-140-3p, previously associated with inflammation, radiation response and tissue damage. In addition, the Bayesian latent variable modeling confirmed significant correlations between these prognostic miRNA signatures and day 14 clinical and functional outcomes from unrelated mice. This study identified plasma miRNA signatures that might be used throughout the latency phase for the prognosis of LRI severity. These results suggest miRNA profiling could be a powerful tool for early LRI diagnosis, thereby improving patient management and treatment outcomes in radiological emergency situations.

**Data availability statement:** All relevant data are within the paper and its Supporting Information files.

**Funding:** This work was supported by IRSN (ASNR since 01/01/2025), Electricité de France EDF (Groupe Gestion Projet Radioprotection, grant V2-303) and the ANR-Astrid program from the Defence Innovation Agency (AID) of the French Armed Forces Ministry grant ANR-22-ASTR-0028 (M.S., M.A.B. and S.F.). The funders had no role in the study design, data collection and analysis, decision to publish, or preparation of the manuscript.

**Competing interests:** The authors do not report any conflict of interests regarding the publication of this paper. A patent was registered (#B254544FR) for the use of the miRNA signatures described in this manuscript for early diagnosis of localized radiation injuries (M.S., M.A.B, and S.F.).

## Introduction

The eventuality of a serious nuclear or radiological event is a major concern considering the ever-growing malevolent threats and current geopolitical situation. In such cases, exposure to high doses of ionizing radiations (IR) can lead to severe health consequences on populations. Therefore, the World Health Organization (WHO) recently highlighted the urgent need for medical countermeasure development, including research about new early diagnostic tools for the medical triage of people exposed to IR during mass accident and radiological emergencies [1]. Among various radiation-induced biological consequences, one of the most severe and frequent consists in radiological burns, also called localized radiation injuries (LRI) [2,3].

LRIs are consecutive to localized exposure to high doses of IR, and are characterized, after an initial and transitory erythema, by a latency phase of variable duration with no overt symptoms, followed by the onset of clinical manifestations [4–7]. After the presymptomatic stage, LRI may present a gradual occurrence of symptoms, with severity and celerity of appearance associated with the radiation dose and the surface of exposed tissue. A second erythema may appear along with edema, followed by the onset of dry to exudative desquamation according to the extent of epidermal destruction, and in most severe cases, by tissue necrosis [4–7]. While clinical manifestations may regress, depending on the exposure severity and medical care effectiveness, late effects of injury may appear months or years after the exposure, which are characterized by unpredictable inflammatory waves with expansion of a fibro-necrotic tissue [8,9]. Treatment recommendations of severe LRI involve a complex combination of surgery and cellular therapy, which effectiveness would benefit from early implementation [9]. Current diagnosis of LRI is usually performed after the appearance of symptoms, and essentially relies on clinical assessment, magnetic resonance imaging, plasma CRP quantification and when available, physical reconstruction of the absorbed radiation dose [2]. Therefore, a new early diagnosis method able to predict the appearance of LRI and its severity before clinical manifestations would be crucial for identification of populations during radiological emergencies. This would improve medical care of patients with enhancement of treatment response and reducing risk of resurgence. Omics approaches allow the identification of relevant molecules (RNA, proteins, metabolites) whose expression levels vary in response to specific stressors. These molecules can be measured in biofluids or tissues and serve as biomarkers of pathophysiological conditions [10].

MicroRNAs (miRNAs) are small non-coding RNA molecules involved in the post-transcriptional regulation of gene expression. Circulating miRNAs are promising candidate biomarkers since they are stable, accessible with minimally invasive blood collection, and easily analyzed with widespread, affordable analytical tools [11]. The usefulness of miRNAs as biomarkers of human pathologies has been demonstrated in a large number of studies, notably in cancers and cardiovascular diseases [12–14]. miRNAs are present both in tissues and blood, and interestingly, changes in circulating miRNA expression have already been correlated with specific tissue injuries [15]. Notably, Jinnin's group described miRNAs as key regulators and serum biomarkers

of severe skin damage resulting from drug induced cutaneous toxicity [16,17]. miRNAs have been identified as circulating biomarkers of whole body irradiation exposure in various preclinical models as well as in patients after radiotherapy [18–21]. In addition, Yadav et al recently described the strong correlation of blood miR-150-5p expression variations with irradiation doses in mice and in patients undergoing myeloablative conditioning irradiation [22]. The results of these studies suggest the potential usefulness of miRNA measurements in complement to current tools dedicated to the management of nuclear and radiological events or radiotherapy treatments, by predicting radiation induced toxicity. Differential miRNA expression can also provide predictive information on organ-specific syndromes after irradiation. For example, early expression of some plasma miRNAs predicts the onset of radiation-induced pneumonia and pulmonary fibrosis in mice [23] and non-human primates [24] after whole-chest irradiation (around 10 Gy). Interestingly, it has been documented that tissue miRNA alterations were associated with acute and late stages of radiation-induced fibrosis in a murine skin model [25]. Moreover, we recently demonstrated the interest of using miRNAs as biomarkers of LRI. Using a preclinical model of hindlimb irradiation, we identified a plasma signature of miRNAs able to discriminate mice according to dose group and injury severity at the manifestation phase [26]. Taken together, this information highlights the potential value of miRNAs as circulating biomarkers for the prognosis of LRI appearance and severity.

Building on our previous study [26], here we aimed to identify early circulating miRNA signature able to predict the appearance and severity of LRI before the onset of clinical manifestations. To this end, we used a similar preclinical model of localized irradiation to perform an integrative approach including clinical, functional and molecular data through adequate biomathematical methods. We conducted broad-spectrum miRNA profiling studies at two different endpoints during latency phase, namely day 1 (early stage) and day 7 (late stage) after irradiation. MiRNA prognostic signatures were further refined using independent animal cohorts for each timepoint.

## Materials and methods

### Experimental model

**Animals.** The experiments were conducted in accordance with French veterinary guidelines and those established by the European Community for the use of experimental animals. The protocols were approved by the institutional animal experimentation and ethics committee (APAFIS Agreement No. 22393-2019101116207862v1) and were conducted following the ARRIVE guidelines (http://arriveguidelines.org). A total of 240 male 8-week-old C57BL/6 mice (Janvier Labs, Le Genest Sainte Isle, France) were used in this study. We applied humane endpoint criteria involving euthanasia of animals when they showed signs of pain, sickness, suffering, or moribund conditions. In particular: loss of body weight, reduced mobility, and hunched body posture were used as main criteria to recognize humane endpoint. No mice died before meeting criteria of euthanasia, nor did they reach endpoint criteria during the course of the study. All animals were euthanized at the end of experiment (*i.e.*, 24h or 7 days post-irradiation) by cervical dislocation following intracardiac blood sampling. The animals were housed in cages (4 mice per cage) within a temperature and humidity-controlled chamber, with alternating 12-hour light and dark cycles and ad libitum access to water and food. Mice were acclimated for one week upon arrival before undergoing experimental procedures. Animal health and behavior were checked daily from arrival to end of experiment.

**Study design.** Four animal cohorts of 60 mice each were used during this study, with two cohorts for day (D) 1 endpoint analysis and two cohorts for D7 analysis (Fig 1A). For each time point (end of experiment), the study was conducted in two phases using two separate animal cohorts. Initially, 60 mice (15 per group) were used for broad-spectrum screening to identify a preliminary miRNA signature capable of distinguishing individuals according to dose group. This preliminary signature was then challenged using an independent cohort of 60 mice (15 per group), by measuring plasma levels of selected miRNAs in order to establish the final miRNA signatures.

**Irradiation & experimental procedure.** Mice hair was removed from both hindlimbs with clippers and depilatory cream. Then, the left hindlimb (foot excepted) was irradiated with X-rays using a medical linear accelerator (Elekta SAS,

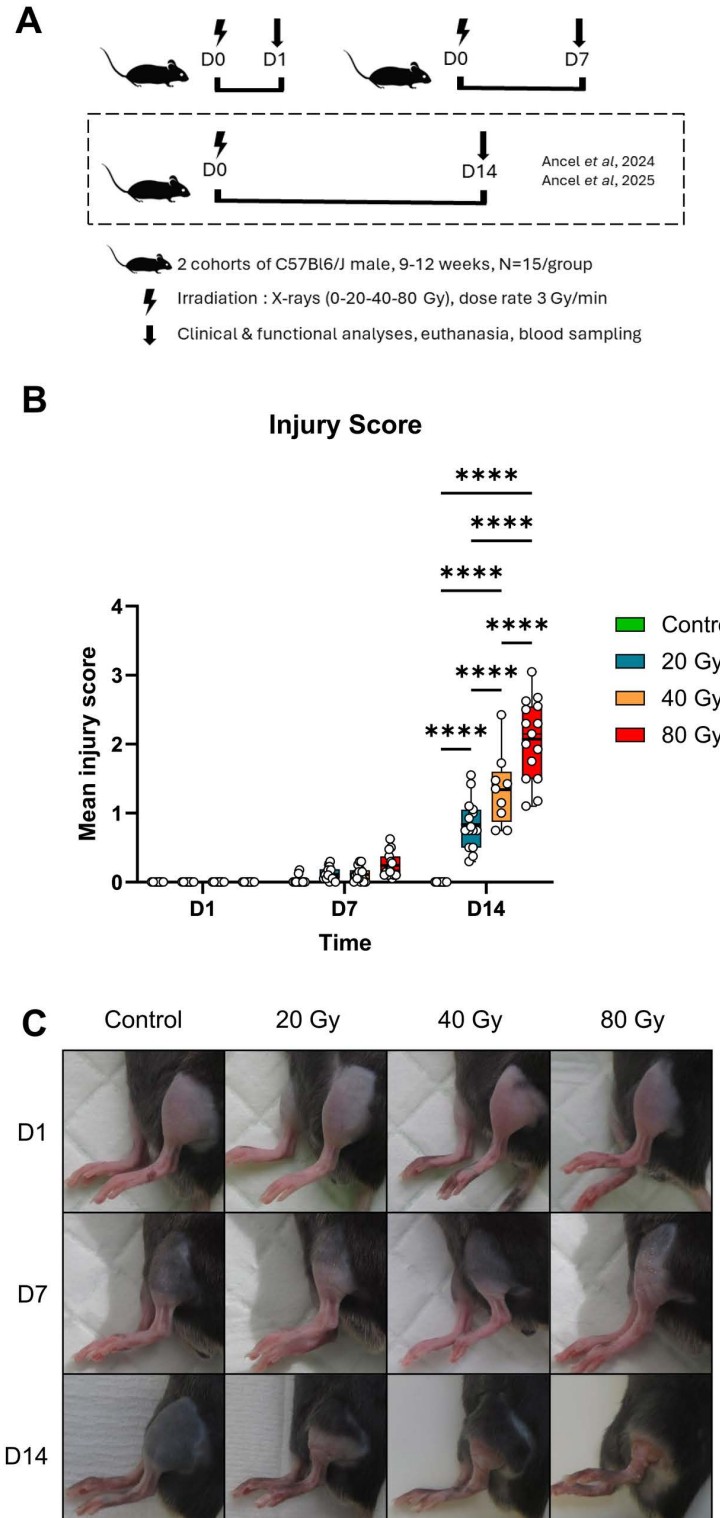

**Fig 1. Pathophysiology of the localized radiation injury at early timepoints.** (A) Experimental design. Two independent mice cohorts were used to perform a broad-spectrum profiling of miRNAs (cohort 1) and then a targeted study (cohort 2) for each time points (D1 and D7, end of experiment). For each cohort, mice were irradiated on their left hindlimb according to 4 irradiation groups with 20, 40 or 80 Gy and 0 for the control group (N=15/group)

with 10 MV X-rays (D0). Clinical and functional analyses were performed at D1 and D7, respectively. Mice were euthanized at D1 or D7 after blood samplings. Data of injury score measured at D14 that are presented in this figure for comparison with those obtained at D1 and D7 were retrieved from our previous studies [26,27]. (B) Mean injury scores assessed for each group at D1 (n = 15/group), D7 (n = 15/group) and D14 (n = 9–15/group) after local irradiation. Each animal is represented. Kruskal-Wallis analysis, * P < 0.05; *** P < 0.001; **** P < 0.0001. (C) Representative photographs of irradiated limbs at D1, D7 and D14 post-irradiation for all dose groups (each photograph corresponds to a distinct animal).

Boulogne-Billancourt, France) with a tension of 10 MV and a dose rate of 3.25 Gy/min under isoflurane anesthesia. Dose rate measurements were taken with an ionization chamber (PTW 31,010–0125 cc) in Kair on a plexiglass plate, with an uncertainty of 5% at k = 2. Mice were randomly assigned to four dose groups and irradiated 5 at a time with single doses of 20, 40, or 80 Gy, while the control group received 0 Gy (N = 15/group). The irradiation field was a rectangle of 2 × 30 cm, and the source-to-sample distance was 1 m. All handling and non-invasive procedures were performed under isoflurane anesthesia, except for weighing, which was carried out before irradiation and prior to euthanasia. One or seven days after irradiation, mice were euthanized by cervical dislocation following intracardiac blood sampling. For each analytical timepoint, this entire procedure was repeated on a second independent cohort (N = 15 animals/group) to refine the miRNA signatures, under the same conditions. Any animal showing signs of skin injury resulting from aggressive behavior within the cages was excluded from all analyses.

## Sample collection

At the end of experiment (D1 or D7 after exposure), collection of 100 µL retro-orbital blood was performed using micropipettes, followed by transfer to BD Microtainer K2 EDTA tubes (BD Biosciences, Le Pont de Claix, France) for analysis of complete blood count (CBC).

A syringe pre-coated with 0.105 M sodium citrate was used to collect around 1 mL of intracardiac blood which was transferred to a 1.5 mL tube containing 75 µL of 0.105 M sodium citrate. The blood samples were processed to obtain platelet-poor plasma (PPP) with two consecutive centrifugations at 1,500 x $g$, 15 min, 4°C and 13,000 x $g$, 2 min, 4°C. PPPs were stored at −80°C until use.

## Clinical approach

**Injury score.** When signs of radiation injury were present (only at D7), an observational injury score was established by two independent experimenters according to a blind analysis, as previously described [26]. For observational score at D14 (see Fig 1), data were retrieved from our previous studies [26,27].

**Cutaneous blood perfusion.** Assessment of limb cutaneous blood perfusion was performed using a Moor LDI2 Laser Doppler Imager (Moor Instruments ltd, Axminster, UK). The measurements lasted for 4 minutes with mice under anesthesia with isoflurane on a heating plate set to 37°C to avoid temperature variation. Analysis of blood perfusion was performed by computing the blood flow ratio between the irradiated limb (IRR) and the non-irradiated, contralateral one (NIR).

**Trans-epidermal water loss.** Trans-epidermal water loss (TEWL) was assessed using a Tewameter® (Courage + Khazaka Electronic GmbH, Köln, Germany). A sensor was placed on the surface of the NIR and IRR limbs for approximately 30 seconds each, which recorded a measurement of water loss (g/h/m²) every second. TEWL data were expressed as the difference of TEWL mean value between IRR and NIR limbs for each animal.

**C-reactive protein (CRP) levels and complete blood count (CBC).** To replicate standard clinical management, we assessed markers of inflammation (CRP levels) and overall health (CBC) at D1 and D7 post-irradiation. CRP quantification was conducted using the Mouse C-Reactive Protein Quantikine ELISA test (R&D Systems, Abingdon, UK) following the manufacturer's protocol, with plasma diluted at 1:4000 using the provided buffer. CBC analysis was carried out on EDTA whole blood using a VetScan HM5 hematology analyzer (Abaxis Europe GmbH, Griesheim, Germany) according to the manufacturer's instructions.

## Molecular approach

**MiRNA extraction and quality control.** Total RNAs were isolated from PPP samples using the miRNeasy® Plasma/ Serum Advanced kit (Qiagen, Courtaboeuf, France) following the manufacturer's instructions. A quality control was performed as previously described [26], using qPCR analysis of three distinct miRNA assays as controls, including a universally expressed miRNA (miR-23a-3p), a miRNA indicative of potential hemolysis (miR-451a), and an exogenous miRNA (cel-miR-39 RNA spike-in control, Qiagen) that was introduced into the sample prior to extraction to monitor the extraction efficiency. The extent of hemolysis in each sample was evaluated by calculating the difference between the expression levels of miR-23a-3p and miR-451a, following a previously established method [28]. Samples that did not meet the quality criteria were excluded from the study.

**Broad-spectrum and targeted qPCR analysis.** Following manufacturer's procedure, reverse transcription was performed with the miRCURY® LNA RT Kit (Qiagen), using a Mastercycler® X50 (Eppendorf). For the broad-spectrum profiling, qPCR was performed using the miRCURY® LNA miRNA miRNome PCR Panels Kit, on Mouse & Rat panel I+II (Qiagen) and the miRCURY® LNA SYBR Green PCR Kit (Qiagen). MiRCURY® LNA miRNA custom PCR panels were used for the targeted analysis. The amplification was run on a Quant Studio 12-PLEX instrument (Applied Biosystems, Foster City, CA) in 10 µL reaction volume for 40 cycles, using 3 µL of cDNA (1:30 dilution), completed with melting curve analysis. Cycle thresholds (Ct) for each assay were determined using automatic baseline with Expression-Suite Software v1.3 (Applied Biosystems), and Ct values>37 were discarded. Delta-Ct values were calculated using adaptive normalization from "MiRAnorm" package in R Software, as described [29]. Briefly, delta-Ct values were obtained by subtracting the assay's Ct values by the mean of an adaptive set of normalizing miRNAs derived from the data through a hierarchical clustering approach based on Euclidean distance and Ward agglomeration method. This set of normalizing miRNAs was obtained by cutting the dendrogram tree from bottom up according to a pre-specified number of three normalizing miRNAs and the most stable cluster minimizing the sum of all distances from its centroid was retained. Normalized (-)delta-Ct values were used for multivariate analysis and generation of miRNA signatures from broad-spectrum and targeted studies.

## Statistical and multivariate analysis

Using GraphPad Prism® v9, a Kruskal-Wallis test (non-parametric) was employed to conduct statistical analyses regarding variations in weight, CBC, CRP levels, injury scores, TEWL and cutaneous blood perfusion.

In order to model multi-scale signatures (miRNAs and clinical data) able to simultaneously identify dose groups and highlight the most significant correlations between different entities of each scale, a supervised multivariate analysis was performed on datasets collected at each time point separately. Two blocks of covariates were defined for each sample: (1) miRNA block with miRNA plasma expression, and (2) Clinical block with CRP, CBC, weight, injury score, TEWL and cutaneous blood perfusion, when available [26].

More precisely, a sparse Multiblock partial least square discriminant analysis (sPLS-DA) model [30] was conducted via the DIABLO framework of the package 'mixOmics' (version 6.18.1) in R software. DIABLO constructed miRNA and clinical block components optimizing their correlation with respect to the dose treatment groups. These components were defined as a linear combination of the original covariates and a Lasso penalization was applied to the loss function of the model to ensure sparsity, variable selection, and interpretability. In all the analyses, leave-one-out cross validation was used to determine the model parameters including number of components and number of features per component.

sPLS-DA DIABLO signature described above was by construction based on correlation with the dose groups and the D1 and D7 clinical data. However, its prognostic ability can be assessed through its capacity to predict a future (post D7) LRI injury. We propose in this study to use clinical D14 data collected in our previous work [26] as a prediction end-point to challenge our prognostic D1-D7 miRNA signatures.

The investigation of the statistical association between the D1 and D7 sPLS-DA coordinates on one hand, and the clinical and functional parameters at D14 on the other hand, is challenging since these two measures were performed on different animal cohorts. This makes the classical statistical approaches inefficient since predictive and response variables are not observed on the same subject. The Bayesian latent variable model [31] offers the possibility to include priors on the distribution of the D1 and D7 sPLS-DA coordinates in the different dose groups to estimate their correlation with the clinical and functional parameters at D14. The model fitting was performed using JAGS software via Markov chain Monte Carlo [32].

## Results

### Pathophysiology of LRI at early timepoints

In our LRI mouse model, mice showed no manifestation of injury at D1 post-irradiation, as shown in Fig 1B and 1C, irrespective of radiation dose. Seven days after irradiation, the majority of mice did not show any sign of injury, except for 4 and 2 mice exposed at 80 and 40 Gy, respectively, which manifested slight erythema and, sometimes, sagging skin appearance, with no significant difference between irradiated and control groups (Fig 1B and 1C). However, at D14 post-irradiation, irradiated mice displayed injuries with severity associated with the dose, as previously shown in our proof-of-concept study [26] ($P < 0.05$).

Cutaneous blood perfusion of irradiated limbs showed no difference between dose groups, or between irradiated groups and controls at D1 post-irradiation (Fig 2A). Similarly, TEWL measurement on both irradiated and non-irradiated limbs did not reveal any difference between groups, indicating no alteration of skin barrier function at that timepoint (Fig 2A). The inability to detect injury with absence of clinical signs and functional impairment agrees with the characteristic presymptomatic, latency phase observed at D1 after irradiation in our preclinical model (Fig 1B and 1C). While plasma CRP level was similar in all groups (Fig.2A), CBC analysis showed a gradual decrease in lymphocyte counts with increasing dose, which was significant for the 80 Gy group ($P < 0.001$ vs control; $P < 0.05$ vs 20 Gy and 40 Gy, Fig 2A), whereas the remaining features of CBC showed no difference between groups (S1 Fig).

At D7 post-irradiation, mice from the 80 Gy group displayed significantly increased blood flow on their irradiated limb when compared to non-irradiated group ($P < 0.01$) (Fig 2B). On the other hand, there was no significant modification of TEWL from irradiated limbs for any group (Fig 2B). Similarly, no difference was observed for plasma CRP level and CBC between groups at that timepoint (Fig 2B and S2 Fig). Hence at D7 post-irradiation, most animals remain in the latency phase of the injury, with a few mice irradiated at 80 Gy starting to show mild erythema and increasing blood perfusion on their irradiated limbs. Apart from skin perfusion analysis, these results are similar to those obtained in our previous study using the same preclinical model of radiological burn with mice monitored on day 7, 10 and 14 post-irradiation [26].

### Broad-spectrum profiling of plasma miRNAs during early and late stage of latency

A broad-spectrum profiling of plasma miRNAs was performed on 750 miRNAs on plasma samples collected at D1 post-irradiation. Expression data of ~300 reproducibly detected miRNAs, along with information from clinical, functional, and biological monitoring were simultaneously analyzed through multivariate analysis (sPLS-DA) in order to classify mice according to their dose group and identify correlations between 2 blocks of variables. The "Macro" block, composed of data from skin perfusion, water loss, CRP, and CBC levels, failed to segregate dose groups (Fig 3A). On the other hand, the "miR" block, composed of miRNA expression levels, allowed the groups segregation at D1 after the exposure. The model identified a panel of 7 stably selected miRNAs (stability scores ≥0.5), which segregated well the control group, the 40 Gy and the 80 Gy groups from each other's, and to a lesser extent the 20 Gy group (Fig 3A). The multivariate analysis was optimized to maximize inter-block correlations, resulting in particular, in correlations between miR-208a-3p levels and WBC (lymphocytes) and between miR-191-5p and CRP levels (Fig 3B).

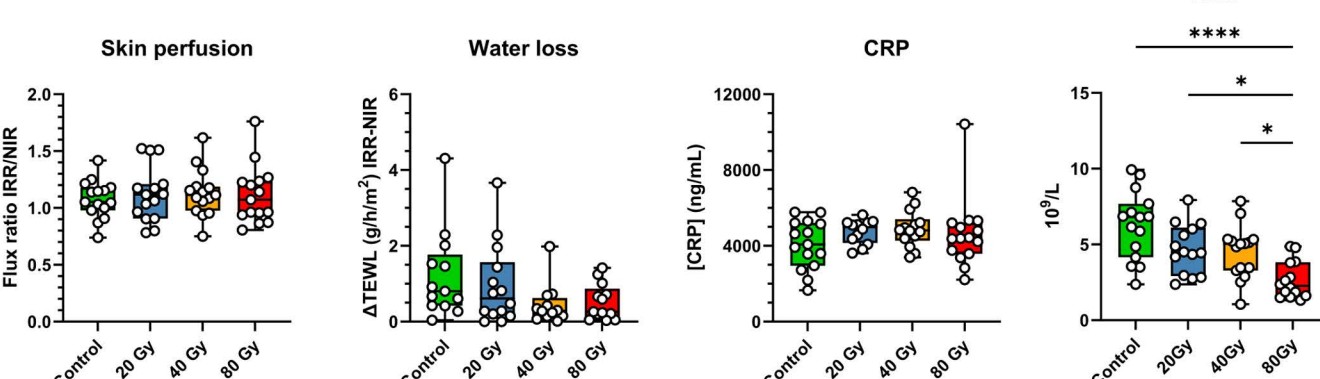

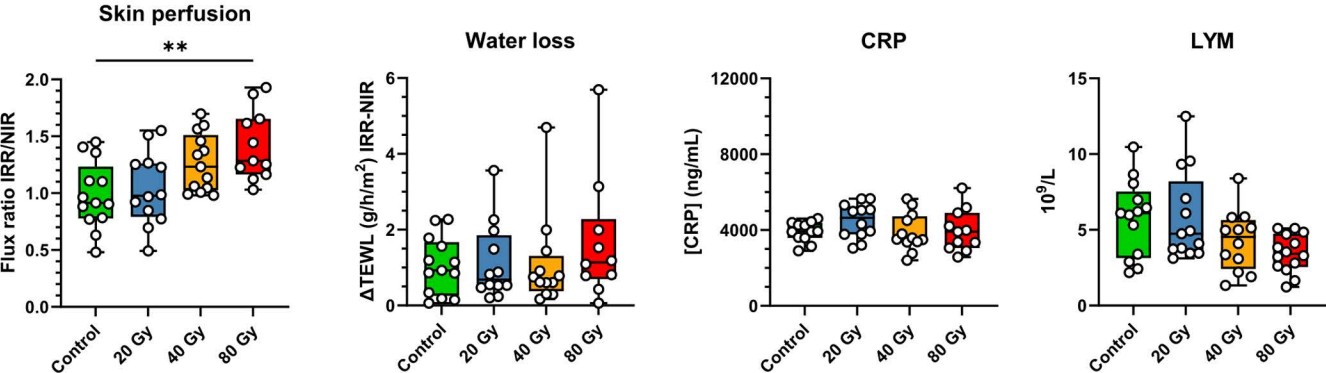

**Fig 2. Clinical and biological monitoring.** Blood-flow ratio of irradiated limb and non-irradiated limb (IRR/NIR) for each group, difference between transepidermal water loss (TEWL) measured on irradiated limb over non-irradiated one (IRR-NIR) for each group (g/h/m²), plasma CRP level (ng/mL) and lymphocyte (LYM) count (10⁹/L) at (A) D1 post-irradiation (n=13–15/group) and (B) D7 post-irradiation (n=12–15/group). Each animal is represented. Kruskal-Wallis analysis, * P<0.05; ** P<0.01; *** P<0.001; **** P<0.0001.

A similar multi-block analysis was performed on an independent cohort at D7 post-irradiation, corresponding to the end of the latency phase, with the "miR" block comprising D7 expression levels of 270 plasma miRNAs and the "Macro" block, composed of D7 functional and biological data including skin perfusion, TEWL, CBC, CRP, and injury score. The multivariate analysis resulted in the identification of a panel of 32 miRNAs allowing group separation, although less efficiently compared to D1 model (Fig 3C). Multi-scale correlations highlighted by the sPLS multi-block analysis were found between different plasma miRNAs and CBC parameters such as lymphocyte counts or clinical features such as skin perfusion (Fig 3D).

### Targeted analysis of plasma miRNAs during early and late stage of latency

Based on the results obtained above from broad-spectrum analyses in D1 and D7 animal cohorts and from the previously published D14 animal cohorts, a selected miRNA panel was designed including the most relevant miRNAs identified

# D1 Post-irradiation

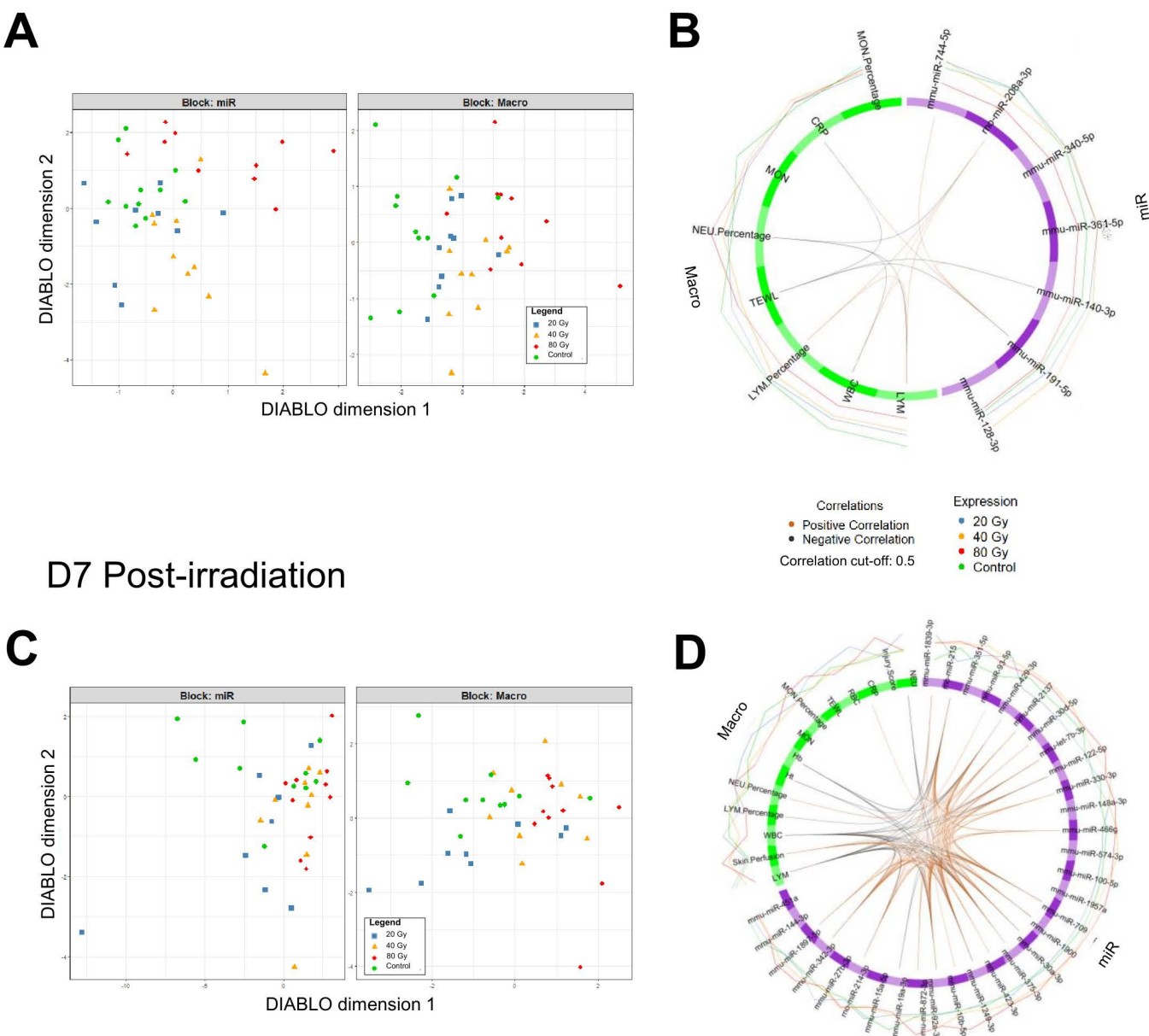

# D7 Post-irradiation

**Fig 3. Broad-spectrum profiling of plasma miRNA at early (D1) and late stage of latency (D7).** (A) Multi-block sPLS-DA scatter plots, with miR block = plasma miRNA expression levels; macro block = cutaneous perfusion, TEWL, CRP, and CBC levels at D1 post-irradiation. (B) Multi-scale correlations between miRNA expression levels and clinical, functional, and biological analyses are illustrated in a circle plot, with positive correlations represented in orange and negative correlations in black, with a cut-off of 0.5. Expression of each parameter as a function of irradiation group is represented with external lines. Control group: n = 10; 20 Gy: n = 9; 40 Gy: n = 10; 80 Gy: n = 10. (C) Multi-block sPLS-DA scatter plots, with miR block = plasma miRNA expression levels; macro block = injury score, cutaneous perfusion, TEWL, CRP, and CBC levels at D7 post-irradiation. (D) Multi-scale correlations between miRNA expression levels and clinical, functional, and biological analyses are illustrated in a circle plot with a cut-off of 0.5. Control group: n = 10; 20 Gy: n = 9; 40 Gy: n = 9; 80 Gy: n = 10.

with highest stability scores and discriminatory power by multivariate analyses. This panel was used to perform targeted miRNA expression analysis from D1 and D7 plasmas of 2 independent cohorts of animals, locally exposed to the same IR doses and monitored for the same biological and functional parameters as above (see Methods).

The DIABLO analysis performed on D1 plasma miRNA expression levels led to the identification of a 14-miRNA signature allowing best achievable segregation of dose groups (Fig 4A). The signature is made up of two dimensions (two sets of miRNAs) which are associated with various aspects of the pathophysiological model. Thus, the DIABLO dimension 1 is the signature component related to the irradiation dose with a mean score significantly different for the 80 Gy group compared to the control group (Fig 4B, $P < 0.05$). On the other hand, the second component contributed to distinguish, though not significantly, the 20 Gy from the 40 Gy groups (DIABLO dimension 2, Fig 4B). Consistent with these results, most miRNAs from the identified signature mainly contributed to the 80 Gy group separation (Fig 4C), while miR-208-3p mainly contributed to the distinction of the control group and miR-324-3p and miR-92a-3p to the 20 Gy group. ROC curves analysis showed the ability of this D1 miRNA signature in segregating each group from the others, with area under the curve (AUC) values ranging from 0.74 to 0.80 with significant $p$-values ($P < 0.05$, Fig 4D). The DIABLO model also determined multiscale correlations between the two blocks and between variables inside each block (Fig 4E). Especially, data from CBC including lymphocyte count were related to several miRNAs. Most miRNAs of the signature were negatively correlated with lymphocyte count, except for miR-208a-3p which displayed a positive correlation with this parameter. Hematocrit, hemoglobin and red blood cell count were also positively correlated with miR-208a-3p, miR-92a-3p and miR-16-5p.

Similarly, the DIABLO multi-block analysis on D7 samples led to the identification of a 16-miRNA panel able to distinguish experimental groups from each other (Fig 5A). The first component of the signature displayed a higher score for the 80 Gy group (Fig 5B), with a significant difference compared to the 40 Gy group ($P < 0.0001$) which indicates the capacity of this component to distinguish between animals exposed to these 2 high doses. On the other hand, the second component showed a mean score statistically lower for the 20 Gy group compared to the control group and the 40 Gy group. Loading plot analysis showed that most miRNAs of the first component displayed a main contribution to the segregation of the 80 and 40 Gy groups (Fig 5C), while miRNAs from the second component accounted evenly for the distinction of all dose groups (Fig 5D). The D7 signature showed great performances for the distinction of groups, especially for the 40 and 80 Gy groups, with AUC values of 0.88 ($P < 0.001$) and 0.93 ($P < 0.001$), respectively (Fig 5E). Notably, the model could also distinguish the control and 20 Gy groups with good performances (AUC > 0.75, $P < 0.05$). Similarly to D1 analysis, the multivariate analysis identified multi-scale correlations between specific miRNAs (e.g. miR-92a-3p, miR-93-5p, miR-19a-3p, miR-186-5p, miR-16-5p, miR-140-3p), skin perfusion and CBC features including WBC, lymphocyte, and neutrophil counts (Fig 5F).

## Early "latency" signatures of localized radiation injury and correlations with expected injury severity

Final components of D1 and D7 signatures are listed in Table 1, together with their stability scores. Interestingly, the two signatures share 9 miRNAs in common (Table 2, with their putative functions according to the literature), with miR-139-5p already being part of our previously identified diagnostic miRNA signature associated with LRI severity at D14 [26]. To go further toward the potential prognostic application of these "latency" signatures, and building on previous demonstration that irradiation dose is strongly correlated with LRI severity in our mouse model [26], we used the Bayesian latent variable model [31] to infer correlations between the present D1 and D7 miRNA signatures and the clinical and functional parameters recorded during manifestation phase at D14 post-irradiation in independent animal cohorts from our previous study (see Materials & Methods) [26]. Results from this modeling revealed interesting significant associations between D1 signature and injury score at D14 post-irradiation ($r = 0.76$, $P < 0.001$), as well as with skin perfusion at D14 ($r = 0.87$, $P < 0.01$, Table 1). Furthermore, the miRNA signature identified at D7 post-irradiation significantly correlated with injury score and skin perfusion at D14 post-irradiation ($p$-values < 0.05).

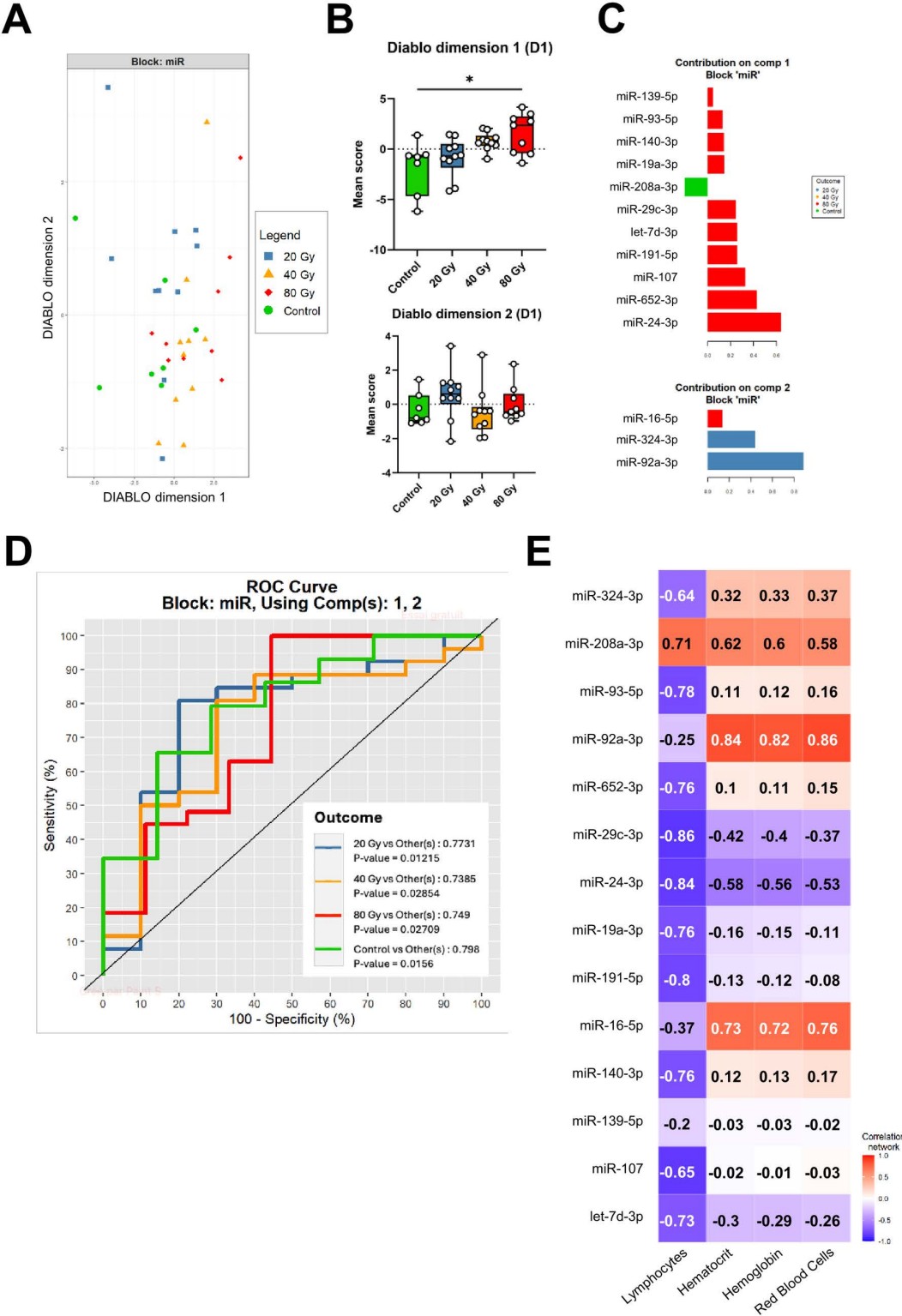

**Fig 4. Targeted analysis of plasma miRNAs at early stage of latency (D1).** (A) Multi-block sPLS-DA scatter plots, with miR block = plasma expression level of 14 miRNAs. (B) Boxplots of scores in each component (DIABLO dimension 1 and DIABLO dimension 2) of the signature according to the dose groups. (C) MiRNAs included in sPLS analysis are plotted with their loading values, representing the main contribution (coefficients in a linear

combination) of the selected original miRNAs. Colours indicate the dose group in which the median expression of each miRNA is maximum. (D) ROC curves demonstrating good performances of the signature for dose group segregation, characterized by AUCs (outcome) > 0.738. (E) Multi-scale correlations between miRNA expression levels and clinical, functional, and biological data are illustrated in a heat-map, with positive correlations represented in red and negative correlations in blue (the stronger the darker). Control group: n = 7; 20 Gy: n = 10; 40 Gy: n = 10; 80 Gy: n = 9.

## Discussion

The absence of an early diagnostic tool for radiation burns delays appropriate medical care [1]. Radiation-induced skin damage vary in severity based on dose and can lead to tissue necrosis [45]. Advances in the field suggest that early treatment would improve therapeutic outcome [9], making it crucial to predict injury appearance and severity before the onset of the first symptoms. Omics approaches hold promise for identifying biomarkers, as molecular changes from irradiation likely precede clinical signs, providing relevant information on the evolution of a pathology [10,46]. MiRNAs have already been documented as being modulated in blood after irradiation in preclinical models of whole-body exposure [18,19,21,47–49], as well as in patient blood after radiotherapy [50,51]. In addition, we previously demonstrated the relevance of plasma miRNAs for the diagnosis of LRI in a mouse preclinical model [26]. They are therefore excellent candidates for the search for minimally or non-invasive early prognosis biomarkers of radiation burns.

Clinical monitoring of mice demonstrated that, in our model, at D1 post-irradiation there was no evidence of injury whether looking at clinical or functional parameters, except for a decrease in circulating lymphocyte count, which could be the result of radiation-induced lymphocyte death or their recruitment in irradiated tissue [52]. However, this unique feature would be insufficient for the detection of irradiated mice since the decrease was only significant for the highest irradiation dose. Likewise, the other CBC parameters, plasma CRP levels and TEWL measurements did not inform about IR exposure for any dose group. At D7 post-irradiation, a minority of mice irradiated at 80 Gy started to show slight erythema, indicating the coming appearance of injury. Of note, these results are in agreement with our previous study where injury signs were not generally detected at D7 and could be observed by D10 post-irradiation [26]. An increase in skin perfusion of irradiated limbs was also observed at D7 for mice exposed to highest doses, which was significant only for the 80 Gy group compared to unexposed controls. However this increase was not found in previous animal cohorts at this timepoint and therefore would not be relevant for LRI early diagnosis considering the variability of this analysis at this stage in our model [26]. In addition, no modification of CBC, CRP levels or TEWL was detected between groups at D7. Therefore, at D1 and D7 there was no obvious sign of injury whether looking at clinical, functional or biological parameters that could effectively identify irradiated mice and estimate putative severity of exposure.

Thus, in our preclinical model, days 1 and 7 post-irradiation corresponded to early and late stages of the latency phase, respectively, and offered the possibility of studying sensitive, non-invasive miRNA biomarkers for the prediction of LRI onset and severity before first clinical manifestations.

Results of the broad-spectrum profiling analysis of plasma miRNAs at D1 post-irradiation led to the identification of a miRNA panel able to distinguish mice according to their dose group. Study of multi-scale correlations between individual values showed relevant correlations between some features of the identified panel and clinical or functional parameters (*e.g.,* lymphocyte counts and miR-208a-3p). This preliminary signature suggests that circulating miRNAs are suitable candidate biomarkers for a very early diagnosis of localized IR exposure. At D7 post-irradiation, although less robust compared to D1, the model obtained from multi-block analysis could identify a group of miRNAs able to segregate dose groups. Consequently, these broad-spectrum D1 and D7 analyses, besides D14 miRNAs signature, led to the selection of a miRNA panel used for the targeted study on 2 additional, independent D1 and D7 mouse cohorts.

Targeted analysis led to the identification of two early signatures, respectively identified at the beginning (D1, 14 miRNAs) and the end (D7, 16 miRNAs) of the presymptomatic latency phase, able to discriminate mice according to their dose group with great performances. Specifically, the signature identified at D1 showed good performances for the distinction of all groups from one another, with AUC values between 0.74 and 0.8 and significant *p*-values (<0.05).

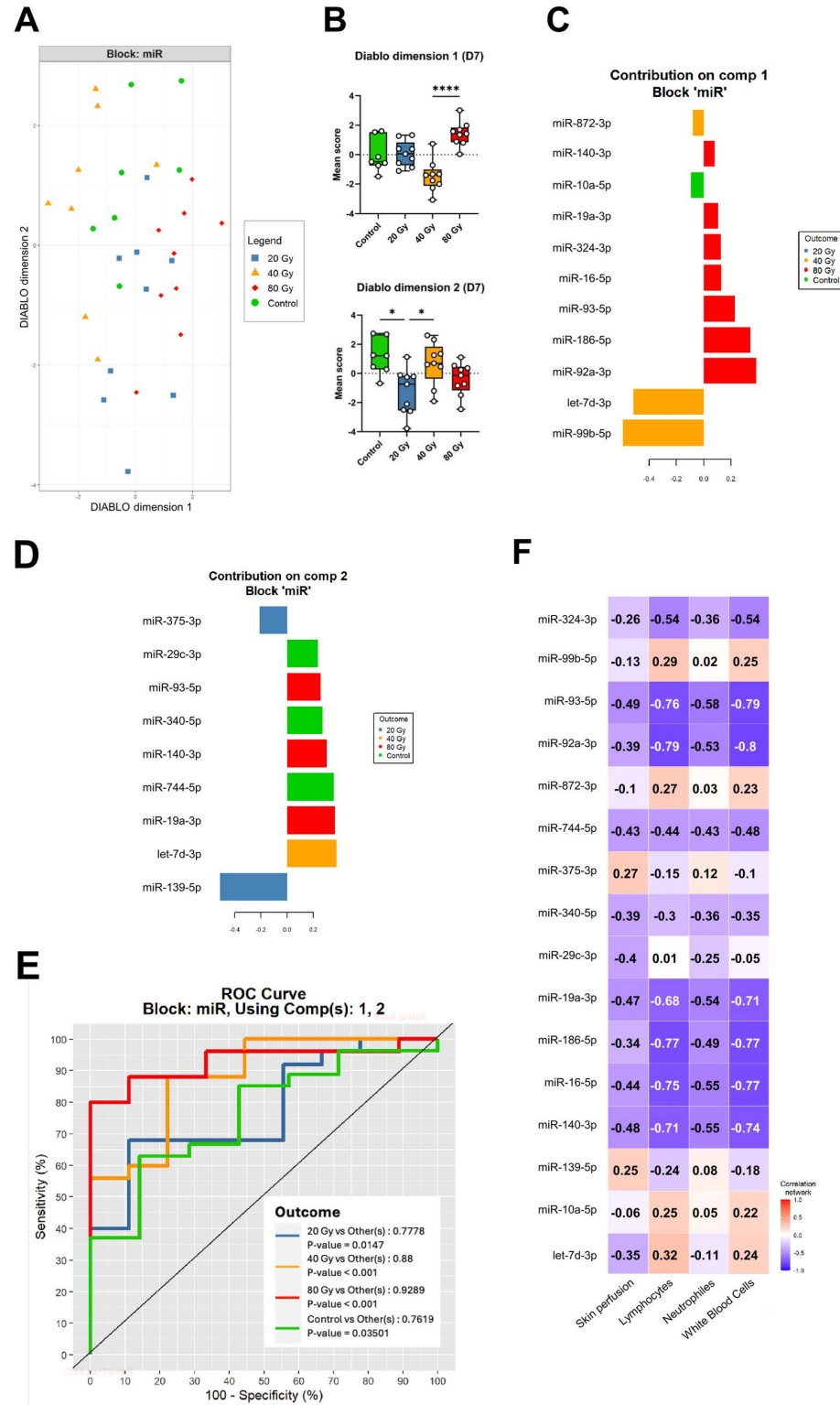

**Fig 5. Targeted analysis of plasma miRNA at late stage of latency (D7).** (A) Multi-block sPLS-DA scatter plots, with miR block = plasma expression level of 16 miRNAs. (B) boxplots of scores in each component (DIABLO dimension 1 and DIABLO dimension 2) of the signature according to dose groups. (C-D) MiRNAs included in sPLS analysis are plotted with their loading values for component 1 (C) and component 2 (D). (E) ROC curves

demonstrating the performances of the signature for dose group segregation. (F) Multi-scale correlations between miRNA expression levels and clinical, functional, and biological data are illustrated in a heat-map, with positive correlations represented in red and negative correlations in blue (the stronger the darker). Control group: n = 7; 20 Gy: n = 9; 40 Gy: n = 9; 80 Gy: n = 9.

**Table 1. Early signatures of localized radiation injury and correlations between early « latency » signatures and expected injury severity.**

| D1 miRNA Signature | | | | | D7 miRNA Signature | | | | |
|---|---|---|---|---|---|---|---|---|---|
| Component | miRNA name | Stability | Correlation with D14 injury score | Correlation with D14 skin perfusion | Component | miRNA name | Stability | Correlation with D14 injury score | Correlation with D14 skin perfusion |
| 1 | let-7d-3p | 1 | $r = 0.76$ $(P < 0.0001)$ | $r = 0.15$ $(P = 0.32)$ | 1 | miR-186-5p | 1 | $r = 0.38$ $(P < 0.05)$ | $r = -0.05$ $(P = 0.76)$ |
| | miR-107 | 1 | | | | miR-92a-3p | 1 | | |
| | miR-191-5p | 1 | | | | miR-99b-5p | 1 | | |
| | miR-19a-3p | 1 | | | | let-7d-3p | 0.88 | | |
| | miR-208a-3p | 1 | | | | miR-93-5p | 0.82 | | |
| | miR-24-3p | 1 | | | | miR-10a-5p | 0.76 | | |
| | miR-29c-3p | 1 | | | | miR-16-5p | 0.74 | | |
| | miR-652-3p | 1 | | | | miR-324-3p | 0.68 | | |
| | miR-93-5p | 1 | | | | miR-19a-3p | 0.65 | | |
| | miR-140-3p | 0.97 | | | | miR-140-3p | 0.62 | | |
| | miR-139-5p | 0.61 | | | | miR-872-3p | 0.62 | | |
| 2 | miR-92a-3p | 0.67 | $r = 0.01$ $(P = 0.93)$ | $r = 0.87$ $(P < 0.01)$ | 2 | miR-139-5p | 0.94 | $r = -0.45$ $(P < 0.05)$ | $r = -0.47$ $(P < 0.05)$ |
| | miR-324-3p | 0.5 | | | | miR-93-5p | 0.82 | | |
| | | | | | | miR-744-5p | 0.79 | | |
| | | | | | | let-7d-3p | 0.68 | | |
| | | | | | | miR-19a-3p | 0.65 | | |
| | | | | | | miR-140-3p | 0.62 | | |
| | | | | | | miR-29c-3p | 0.58 | | |
| | | | | | | miR-375-3p | 0.35 | | |
| | | | | | | miR-340-5p | 0.32 | | |

**Table 2. Common miRNAs of D1 and D7 signatures with their putative functions.**

| miRNA | Putative function | References |
|---|---|---|
| let-7d-3p | Potential inflammation biomarker in human subjects | [33] |
| miR-139-5p | Regulation of inflammatory skin wound in mice | [34] |
| miR-140-3p | Deregulated in bone marrow derived extracellular vesicles 24h after irradiation; Deregulated after gamma and X irradiation (chronic low dose rate, 3 Gy) in mice liver | [35,36] |
| miR-16-5p | Inhibition of which led to significant reduction of inflammation and DNA double strand breaks in microvessel cells after space radiation simulation | [37] |
| miR-19a-3p | Upregulation in rat blood 2 weeks after irradiation (whole thorax irradiation X rays) | [38] |
| miR-29c-3p | Upregulate melanogenesis after UVB-irradiation of keratinocytes | [39] |
| miR-324-3p | Related to inflammation and muscle inflammatory response | [40,41] |
| miR-92a-3p | Upregulated in extracellular vesicles from healthy donors irradiated plasma (2Gy) | [42] |
| miR-93-5p | Improves radiosensitivity in breast cancer cells; Involved in inflammatory processes in murine skin lesion following UVB irradiation | [43,44] |

Furthermore, this signature displayed a component with a particular contribution to the 80 Gy group segregation, characterized by the fall of lymphocyte counts at D1 post-irradiation. This was confirmed by the significant correlations identified between miRNA plasma expression levels and CBC parameters at D1 post-irradiation which revealed interesting associations between some miRNAs and lymphocyte levels, suggesting involvement of those miRNAs in inflammatory processes. Indeed, some miRNAs have already been identified in literature as inflammation biomarkers (let-7d-3p [33]) or key regulators (miR-324-3p [41], miR-93-5p [44]). Similarly, the identified D7 miRNA signature comprised two components which contributed more specifically to the separation of the higher dose groups (component 1) and of the lower dose groups (component 2). This 16-miRNA signature showed great performances for the segregation of each dose group from the others with AUC values ranging from 0.93 ($P < 0.01$) for the 80 Gy group to 0.76 ($P < 0.05$) for the control group. Seven days after irradiation some miRNA expression levels were still significantly related to lymphocyte and neutrophile counts, which suggests again a potential role of these miRNAs in inflammatory processes. Interestingly, some of those miRNAs were also related to lymphocyte counts at D1 post-irradiation (miR-324-3p, miR-93-5p, miR-140-3p) which suggests a potential interest in these miRNAs for the follow-up of inflammation during the first week after irradiation in this model.

These results demonstrate the relevance of using plasma miRNAs as early biomarkers of LRI. Indeed, we already showed that LRI severity increases with irradiation dose, which suggests that early, dose-associated miRNA signatures should also be related to expected injury severity. This point is supported by the Bayesian latent variable model which correlated the D1 and D7 signatures with injury scores and skin perfusion measured at D14 post-irradiation (manifestation phase) from our previous study [26]. Furthermore, those two signatures share 9 common miRNAs (i.e., a majority within each signature), which paves the way for determining a single relevant signature able to predict LRI severity with samples collected during the latency phase. Eventually, an ideal signature would be informative during both LRI latency and manifestation phases, providing clue about severity, kinetics of injury development, healing potential and risk of resurgences. However, despite the fact that injury severity and irradiation dose are strongly correlated in our LRI model, they are not strictly overlapping. Indeed, we previously showed that mice irradiated at 40 Gy can display slight to severe injuries similar to animals from the 20 or 80 Gy group, depending on each animal and on the analyzed timepoint [26]. Further studies looking at early miRNA biomarkers during the latency phase that would relate to injury severity at later timepoints in the same animal should improve the performances of the signatures described in the present paper for prognosis of LRI appearance and severity.

Among the 9 common miRNAs of the two signatures, miR-139-5p was also part of the diagnostic plasma signature previously identified at D14 post-irradiation (manifestation phase) [26]. This miRNA was also already identified as a regulator of inflammatory response in a model of bacteria infected skin wound healing [34]. Interestingly, a majority of those 9 miRNAs have previously been identified as being modulated in response to IR in several preclinical studies. Notably, miR-19a-3p was found upregulated in rat blood 2 weeks after whole thorax irradiation using X rays [38], while miR-140-3p was shown deregulated in extracellular vesicles derived from bone marrow 24h after irradiation [36] and in mice liver after chronic, low-dose rate gamma and X irradiation [35]. In addition, miR-92a-3p was deregulated in exosomes from irradiated plasma of healthy donors [42]. Furthermore, a recent study showed that antagomir-mediated inhibition of miR-16-5p led to significant reduction of inflammation and DNA double strand breaks in human microvascular cells exposed to simulated space radiation conditions [37]. Other miRNAs are related to inflammation, including let-7d-3p which is a potential inflammation biomarker in human subjects [33] and miR-324-3p which is known to be involved in inflammation processes and notably in muscular inflammatory conditions [40,41]. Finally, miR-29c-3p has been discovered as an up-regulator of melanogenesis after UVB-irradiation in keratinocytes [39]. All this information is interesting when considering that inflammation, and skin and muscle damage are part of LRI pathophysiology and therefore support the relevance of signatures whose components could be part of molecular alterations.

## Conclusions

In conclusion, this study provided the first plasma miRNA signature as early biomarkers of LRI, able to segregate animals according to dose groups during the presymptomatic, latency phase. Two signatures were identified at early and late stages of latency, sharing a majority of miRNAs known to be involved in molecular pathways of LRI pathophysiology. These results pave the way for the identification of a unique plasma miRNA signature able to predict LRI onset and severity during the whole latency phase, thereby contributing to enriching the diagnostic arsenal of LRI. Such molecular tool would be crucial for medical triage during a nuclear or radiological event and would improve medical care for patients. However, the robustness of such signature must be challenged using other preclinical models including large animals, as well as on female subjects.

## Supporting information

**S1 Fig. CBC at D1 post-irradiation.** Each animal is represented (n = 13–15/group). Kruskal-Wallis analysis, * $P < 0.05$; *** $P < 0.001$; **** $P < 0.0001$. RBC, red blood cells; Hb, hemoglobin; Ht, hematocrit; PLT, platelet; WBC, white blood cells; NEU, neutrophils; LYM, lymphocytes; MCHC, Mean corpuscular hemoglobin concentration; MCH, mean corpuscular hemoglobin; MONO, monocytes.
(TIF)

**S2 Fig. CBC at D7 post-irradiation.** Each animal is represented (n = 12–15/group). Kruskal-Wallis analysis.
(TIF)

**S3 Table. Raw data for each cohort of the study.**
(XLSX)

**S4 Table. Raw data of QC miRNAs and controls for each cohort.**
(XLSX)

**S5 Table. Injury scores and laser Doppler skin perfusion ratios (IRR/NIR) at day-14 (from ref26).**
(XLSX)

## Acknowledgments

The authors wish to thank D. Denais-Lalieve, F. Voyer, R. Granger and A. Sache, for assistance with animal housing and handling. The authors thank M. Razanajatovo and Y. Ristic for their help with irradiation on the Elekta Synergy Platform. The authors wish to thank Emmanuelle Guillot-Combe for her support and advice.

## Author contributions

**Conceptualization:** Radia Tamarat, Maâmar Souidi, Mohamed Amine Benadjaoud, Stéphane Flamant.

**Data curation:** Lucie Ancel, Guillaume Thoër, Mohamed Amine Benadjaoud, Stéphane Flamant.

**Formal analysis:** Lucie Ancel, Jules Gueguen, Guillaume Thoër, Jules Marçais, Aïda Chemloul, Maâmar Souidi, Mohamed Amine Benadjaoud, Stéphane Flamant.

**Funding acquisition:** Bernard Le Guen, Marc Benderitter, Maâmar Souidi, Mohamed Amine Benadjaoud, Stéphane Flamant.

**Investigation:** Lucie Ancel, Maâmar Souidi, Mohamed Amine Benadjaoud, Stéphane Flamant.

**Methodology:** Lucie Ancel, Guillaume Thoër, Aïda Chemloul, Mohamed Amine Benadjaoud, Stéphane Flamant.

**Software:** Jules Marçais, Mohamed Amine Benadjaoud.

**Supervision:** Maâmar Souidi, Mohamed Amine Benadjaoud, Stéphane Flamant.

**Validation:** Marc Benderitter, Radia Tamarat, Maâmar Souidi, Mohamed Amine Benadjaoud, Stéphane Flamant.

**Writing – original draft:** Lucie Ancel.

**Writing – review & editing:** Maâmar Souidi, Mohamed Amine Benadjaoud, Stéphane Flamant.

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
