## [Decision Letter · Decision Letter 0]

3 Jun 2025

Dear Dr. FLAMANT,

Thank you for submitting your manuscript to PLOS ONE. After careful consideration, we feel that it has merit but does not fully meet PLOS ONE’s publication criteria as it currently stands. Therefore, we invite you to submit a revised version of the manuscript that addresses the points raised during the review process.

We look forward to receiving your revised manuscript.

Kind regards,

Dr. Daniel X Zhang

Academic Editor

PLOS ONE

2. Thank you for your submission to PLOS ONE. We note that your study design may include death of a regulated animal as a likely outcome or planned experimental endpoint. At this time, we request that you please report additional details in your Methods section regarding animal care and use for the survival study, as per our editorial guidelines (http://journals.plos.org/plosone/s/submission-guidelines#loc-humane-endpoints).     

For easy reference, we have attached a checklist that may be relevant for your submission. Please complete all items on the checklist at the following link:   http://journals.plos.org/plosone/s/file?id=bb1d/plos-one-humane-endpoints-checklist.docx        

Please upload the completed checklist as file type “Other” when resubmitting your manuscript. This document is for internal journal use only and will not be published if your article is accepted. We very much appreciate your attention to these requests and support of improved reporting standards in PLOS ONE submissions.

Reviewers' comments:

Reviewer's Responses to Questions

**Comments to the Author**

1. Is the manuscript technically sound, and do the data support the conclusions?

Reviewer #1: Yes

Reviewer #2: Yes

2. Has the statistical analysis been performed appropriately and rigorously?

Reviewer #1: No

Reviewer #2: Yes

3. Have the authors made all data underlying the findings in their manuscript fully available?

Reviewer #1: No

Reviewer #2: Yes

4. Is the manuscript presented in an intelligible fashion and written in standard English?

Reviewer #1: Yes

Reviewer #2: Yes

Reviewer #1: This is an important study that should be published so that others can build on it. Experiments were done thoroughly. Data analysis with miRNA is challenging and the robustness of the authors conclusions will only be determined over time as others build on it. The results are interesting but not convincing. To support that, the following needs to be clarified.

1. Which samples were rejected (hemolysis, etc.) based on their various quality criteria should be noted. For instance, in the D1 broad spectrum, only 40 of the 60 animals are reported.

2. The qPCR controls should be reported.

3. qPCR standard curves for the kits used should be reported

4. Were there any normalization steps taken? This should be clearly stated and described.

5. In the prognostic model, what were the D14 scores? It should also be in the supplemental data.

Reviewer #2: Title: Presymptomatic microRNA-based biomarker signatures for the prognosis of localized radiation injury in mice

Author(s): Lucie Ancel, Jules Gueguen, Guillaume Thoer, Jules Marçais, Aïda Chemloul, Bernard Le Guen, Marc Benderitter, Radia Tamarat, Maâmar Souidi, Mohamed Amine Benadjaoud, Stéphane Flamant

Reviewer´s comments:

The presented article by authors Ancel et al. aimed to identify plasma miRNA signatures that might be used for the prognosis of localized radiation injury severity before symptoms appear. The authors already have experience in this field, referring to their previous publications (https://doi.org/10.1038/s41598-024-52258-2;
https://doi.org/10.1038/s41598-025-85717-5) that are properly cited in this manuscript. To obtain study results, the authors used suitable methodologies as well as statistical analyses. I appreciate that the identified candidate miRNAs were refined using two independent mouse cohorts.

Early diagnosis of localized radiation injury as well as setting up the appropriate treatment are hampered due to its presymptomatic phase. Therefore, research in the area of potential biomarkers for early prediction of radiation injury severity is very welcomed. New knowledge in this field could help to improve patient management and treatment outcomes in radiological emergency situations. This is a well-written manuscript containing promising results that point to the possible role of selected miRNAs in predicting the severity of localized radiation injury. All presented data are appropriately discussed and graphically illustrated. However, the quality of all figures is very bad and needs to be improved, some figures were not even readable.

I have the following recommendations to improve the manuscript:

1. As I already mentioned, the quality of all figures needs to be improved.

2. Line 91: The authors did not explain the abbreviation NHPs, this should be fixed.

Based on previous comments, I recommend this article to be accepted for publication after minor revision in this journal.

**Do you want your identity to be public for this peer review?** For information about this choice, including consent withdrawal, please see our Privacy Policy

Reviewer #1: No

Reviewer #2: No

---

## [Author Response · Author response to Decision Letter 1]

30 Jun 2025

Point-by-point response to the reviewers’ comments

Dear Editor and reviewers,

We are pleased to produce below a point-by-point response to reviewer’s comments. We thank the reviewers for their thorough analysis of our work and their comments and remarks, which led to modifications of the manuscript that should allow a better understanding of our study.

In particular we clarified the discrepancy between the total number of animals involved in the study and samples used for multivariate analyses. We also addressed the comments regarding the analysis of qPCR data by adding more details on the adaptive normalization procedure and providing the supporting S4 Table with quality check data.

In addition, we corrected the column headings with erroneous labels in Table 1 and provide supporting D14 injury scores in the supplementary S5 Table.

The modifications of the text are highlighted in yellow in this revised version.

Reviewer #1:

This is an important study that should be published so that others can build on it. Experiments were done thoroughly. Data analysis with miRNA is challenging and the robustness of the authors conclusions will only be determined over time as others build on it. The results are interesting but not convincing. To support that, the following needs to be clarified.

1. Which samples were rejected (hemolysis, etc.) based on their various quality criteria should be noted. For instance, in the D1 broad spectrum, only 40 of the 60 animals are reported.

A total of 60 animals (15 per group) were included in each study. This number was chosen based on the team's prior experience, with the aim of obtaining plasma samples of up to 10 animals per group for molecular analyses.

It is well established from previous studies that some animals or samples may be excluded from the study due to various factors. In particular, any animal showing signs of skin injury resulting from aggressive behavior within the cages was excluded from all analyses. This was done to avoid introducing bias when evaluating skin-muscle injury biomarkers. Additional common reasons for exclusion included inadequate blood sample quality, such as insufficient volume, hemolysis, or suspected RNA degradation. Moreover, parts of plasma volume were allocated to metabolomics studies (paper in preparation), similar to our previous study at D14 (Ancel 2025, ref 27).

However, these sample quality issues did not compromise the reliability of the clinical assessments and functional monitoring conducted during the animals’ lifetime, including injury score, cutaneous blood perfusion, transepidermal water loss, and complete blood counts (CBC), the latter of which was performed using an independent retro-orbital blood sampling method.

As a result, the number of animals reported in clinical and functional follow-up may differ from those included in molecular analyses. Only animals with high-quality data across all molecular, clinical and functional endpoints were retained for multivariate analyses. In any case, exposure to RI or dose group were not specifically associated with sample exclusion decision.

Given this framework, the target number for molecular analyses was set to 6-10 animals per group, even if a higher number of animals yielded blood samples of sufficient quality. Any surplus samples were stored for potential future analyses. Nevertheless, we chose to present all available data related to clinical and functional assessments (Fig.1 & 2) in order to highlight the robustness and consistency of our preclinical LRI model.

2. The qPCR controls should be reported.

As requested, qPCR Ct values for spike-in and controls for the samples used in multivariate analysis are now provided as supplementary file S4 Table.

3. qPCR standard curves for the kits used should be reported.

In this study, we used the complete miRCURY solution (Qiagen) to perform the RT-qPCR experiments. RT reaction was performed using the miRCURY LNA RT kit. The qPCR reactions were conducted using the 2x miRCURY LNA SYBR Green PCR Master Mix, first on the miRCURY LNA miRNA Mouse & Rat miRNome panels I+II (for broad spectrum analyses), then on miRCURY LNA miRNA custom panels (for targeted analyses) designed according to Qiagen GeneGlobe recommendations.

All these PCR panels used proprietary LNA-enhanced primers for miRNA amplification that were specifically designed in order to amplify miRNAs under the same experimental conditions and with high discriminatory power between closely related miRNA sequences. Similar to our previously published study (Ancel 2024, ref 26), we considered that these LNA miRNA assays were fully validated and optimized and that the manufacturer provided satisfactory information regarding assay parameters including detection range and standard curve (as shown on their website at https://www.qiagen.com/us/products/discovery-and-translational-research/pcr-qpcr-dpcr/qpcr-assays-and-instruments/mirna-qpcr-assay-and-panels/mircury-lna-mirna-pcr-assays). In addition, we used the reagents recommended by the manufacturer (i.e. 2x miRCURY LNA SYBR Green PCR Master Mix) to optimize amplification performances. We used the -ΔCt values obtained following data normalization with the MiRAnorm package (see answer to comment #4 below) to perform the multivariate analyses described in this study.

4. Were there any normalization steps taken? This should be clearly stated and described.

As we mentioned in the M&M section of the manuscript, the ΔCt values were obtained through an adaptive normalization combining clustering and selection of the most stable cluster. As suggested by the reviewer, we improved the revised manuscript by describing the approach in more details with additional technical information (p10, line 214):

“Delta-Ct values were calculated using adaptive normalization from “MiRAnorm” package in R Software, as described [29]. Briefly, Delta-Ct values were obtained by subtracting the assay’s Ct values by the mean of an adaptive set of normalizing miRNAs derived from the data through a hierarchical clustering approach based on Euclidean distance and Ward agglomeration method. This set of normalizing miRNAs was obtained by cutting the dendrogram tree from bottom up according to a pre-specified number of three normalizing miRNAs and the most stable cluster minimizing the sum of all distances from its centroid was retained.”

5. In the prognostic model, what were the D14 scores? It should also be in the supplemental data.

We apologize for the mislabeling of the column headings corresponding to miRNA name and stability score in Table 1. We modified the Table 1 accordingly, with modifications highlighted in yellow. Moreover, as suggested by the reviewer, we now report the D14 injury scores in the supplementary document S5 Table. These values correspond to animals from our previous study (Ancel 2024, ref 26).

Reviewer #2:

The presented article by authors Ancel et al. aimed to identify plasma miRNA signatures that might be used for the prognosis of localized radiation injury severity before symptoms appear. The authors already have experience in this field, referring to their previous publications (https://doi.org/10.1038/s41598-024-52258-2;
https://doi.org/10.1038/s41598-025-85717-5) that are properly cited in this manuscript. To obtain study results, the authors used suitable methodologies as well as statistical analyses. I appreciate that the identified candidate miRNAs were refined using two independent mouse cohorts.

Early diagnosis of localized radiation injury as well as setting up the appropriate treatment are hampered due to its presymptomatic phase. Therefore, research in the area of potential biomarkers for early prediction of radiation injury severity is very welcomed. New knowledge in this field could help to improve patient management and treatment outcomes in radiological emergency situations. This is a well-written manuscript containing promising results that point to the possible role of selected miRNAs in predicting the severity of localized radiation injury. All presented data are appropriately discussed and graphically illustrated. However, the quality of all figures is very bad and needs to be improved, some figures were not even readable.

I have the following recommendations to improve the manuscript:

1. As I already mentioned, the quality of all figures needs to be improved.

Thank you for your favorable comment. As a matter of fact, we were also surprised at first when we downloaded the compiled pdf document automatically prepared by the PLOS One submission system and we saw the poor quality of the figures. In fact, at the top right of each figure page in the pdf document, there is a link to download the original .tif figure, so you can read the details and the writing in high quality.

2. Line 91: The authors did not explain the abbreviation NHPs, this should be fixed.

The modification was made.

---

## [Decision Letter · Decision Letter 1]

22 Jul 2025

Presymptomatic microRNA-based biomarker signatures for the prognosis of localized radiation injury in mice

PONE-D-25-26068R1

Dear Dr. FLAMANT,

We’re pleased to inform you that your manuscript has been judged scientifically suitable for publication and will be formally accepted for publication once it meets all outstanding technical requirements.

Kind regards,

Daniel X Zhang

Academic Editor

PLOS ONE

Additional Editor Comments (optional):

Reviewers' comments:

Reviewer's Responses to Questions

**Comments to the Author**

Reviewer #1: All comments have been addressed

Reviewer #2: All comments have been addressed

2. Is the manuscript technically sound, and do the data support the conclusions?

Reviewer #1: Yes

Reviewer #2: Yes

3. Has the statistical analysis been performed appropriately and rigorously?

Reviewer #1: Yes

Reviewer #2: Yes

4. Have the authors made all data underlying the findings in their manuscript fully available?

Reviewer #1: Yes

Reviewer #2: Yes

5. Is the manuscript presented in an intelligible fashion and written in standard English?

Reviewer #1: Yes

Reviewer #2: Yes

Reviewer #1: (No Response)

Reviewer #2: The Authors adressed all my comments I had in my previous review report. I have no further requirements. Based on this statement I recommend accepting the manuscript it in its current form.

**Do you want your identity to be public for this peer review?** For information about this choice, including consent withdrawal, please see our Privacy Policy

Reviewer #1: No

Reviewer #2: No

---

## [Editor Report · Acceptance letter]

PONE-D-25-26068R1

PLOS ONE

Dear Dr. FLAMANT,

I'm pleased to inform you that your manuscript has been deemed suitable for publication in PLOS ONE. Congratulations! Your manuscript is now being handed over to our production team.

Kind regards,

on behalf of

Dr. Daniel X Zhang

Academic Editor

PLOS ONE